# ENTITY DIVIDER WITH LANGUAGE GROUNDING IN MULTI-AGENT REINFORCEMENT LEARNING

## ABSTRACT

We investigate the use of natural language to drive the generalization of policies in multi-agent settings. Unlike single-agent settings, the generalization of policies should also consider the influence of other agents. Besides, with the increasing number of entities in multi-agent settings, more agent-entity interactions are needed for language grounding, and the enormous search space could impede the learning process. Moreover, given a simple general instruction, *e.g., beating all enemies*, agents are required to decompose it into multiple subgoals and figure out the right one to focus on. Inspired by previous work, we try to address these issues at the entity level and propose a novel framework for language grounding in multi-agent reinforcement learning, *entity divider* (EnDi). EnDi enables agents to independently learn subgoal division at the entity level and act in the environment based on the associated entities. The subgoal division is regularized by agent modeling to avoid subgoal conflicts and promote coordinated strategies. Empirically, EnDi demonstrates the strong generalization ability to unseen games with new dynamics and expresses the superiority over existing methods.

## 1 INTRODUCTION

The generalization of reinforcement learning (RL) agents to new environments is challenging, even to environments slightly different from those seen during training (Finn et al., 2017). Recently, language grounding has been proven to be an effective way to grant RL agents the generalization ability (Zhong et al., 2019; Hanjie et al., 2021). By relating the dynamics of the environment with the text manual specifying the environment dynamics at the entity level, the language-based agent can adapt to new settings with unseen entities or dynamics. In addition, language-based RL provides a framework for enabling agents to reach user-specified goal states described by natural language (Küttler et al., 2020; Tellex et al., 2020; Branavan et al., 2012). Language description can express abstract goals as sets of constraints on the states and drive generalization.

However, in multi-agent settings, things could be different. First, the policies of others also affect the dynamics of the environment, while text manual does not provide such information. Therefore, the generalization of policies should also consider the influence of others. Second, with the increasing number of entities in multi-agent settings, so is the number of agent-entity interactions needed for language grounding. The enormous search space could impede the learning process. Third, sometimes it is unrealistic to give detailed instructions to tell exactly what to do for each agent. On the contrary, a simple goal instruction, *e.g., beating all enemies* or *collecting all the treasures*, is more convenient and effective. Therefore, learning subgoal division and cultivating coordinated strategies based on one single general instruction is required.

The key to generalization in previous works (Zhong et al., 2019; Hanjie et al., 2021) is grounding language to dynamics at the entity level. By doing so, agents can reason over the dynamic rules of all the entities in the environment. Since the dynamic of the entity is the basic component of the dynamics of the environment, such language grounding is invariant to a new distribution of dynamics or tasks, making the generalization more reliable. Inspired by this, in multi-agent settings, the influence of policies of others should also be reflected at the entity level for better generalization. In addition, after jointly grounding the text manual and the language-based goal (goal instruction) to environment entities, each entity has been associated with explicit dynamic rules and relationships

with the goal state. Thus, the entities with language grounding can also be utilized to form better strategies.

We present two goal-based multi-agent environments based on two previous single-agent settings, *i.e.,* MESSENGER (Hanjie et al., 2021) and RTFM (Zhong et al., 2019), which require generalization to new dynamics (*i.e.,* how entities behave), entity references, and partially observable environments. Agents are given a document that specifies environment dynamics and a language-based goal. Note that one goal may contain multiple subgoals. In more detail, in multi-agent messenger, agents are required to bring all the messages to the targets, while in multi-agent RTFM, the general goal is to eliminate all monsters in a given team. Thus, one single agent may struggle or be unable to finish it. In particular, after identifying relevant information in the language descriptions, agents need to decompose the general goal into many subgoals and figure out the optimal subgoal division strategy. Note that we focus on interactive environments that are easily converted to symbolic representations, instead of raw visual observations, for efficiency, interpretability, and emphasis on abstractions over perception.

In this paper, we propose a novel framework for language grounding in multi-agent reinforcement learning (MARL), ***entity divider*** (EnDi), to enable agents *independently* learn subgoal division strategies at the entity level. Specifically, each EnDi agent first generates a language-based representation for the environment and decomposes the goal instruction into two subgoals: *self* and *others*. Note that the subgoal can be described at the entity level since language descriptions have given the explicit relationship between the goal and all entities. Then, the EnDi agent acts in the environment based on the associated entities of the *self* subgoal. To consider the influence of others, the EnDi agent has two policy heads. One is to interact with the environment, and another is for agent modeling. The EnDi agent is jointly trained end-to-end using reinforcement learning and supervised learning for two policy heads, respectively. The gradient signal of the supervised learning from the agent modeling is used to regularize the subgoal division of *others*.

Our framework is the first attempt to address the challenges of grounding language for generalization to unseen dynamics in multi-agent settings. EnDi can be instantiated on many existing language grounding modules and is currently built and evaluated in two multi-agent environments mentioned above. Empirically, we demonstrate that EnDi outperforms existing language-based methods in all tasks by a large margin. Importantly, EnDi also expresses the best generalization ability on unseen games, *i.e.,* zero-shot transfer. By ablation studies, we verify the effectiveness of each component, and EnDi indeed can obtain coordinated subgoal division strategies by agent modeling. We also argue that many language grounding problems can be addressed at the entity level.

## 2  RELATED WORK

**Language grounded policy-learning.** Language grounding refers to learning the meaning of natural language units, *e.g.,* utterances, phrases, or words, by leveraging the non-linguistic context. In many previous works (Wang et al., 2019; Blukis et al., 2019; Janner et al., 2018; Küttler et al., 2020; Tellex et al., 2020; Branavan et al., 2012), the text conveys the goal or instruction to the agent, and the agent produces behaviors in response after the language grounding. Thus, it encourages a strong connection between the given instruction and the policy.

More recently, many works have extensively explored the generalization from many different perspectives. Hill et al. (2020a; 2019; 2020b) investigated the generalization regarding novel entity combinations, from synthetic template commands to natural instructions given by humans and the number of objects. Choi et al. (2021) proposed a language-guided policy learning algorithm, enabling learning new tasks quickly with language corrections. In addition, Co-Reyes et al. (2018) proposed to guide policies by language to generalize on new tasks by meta learning. Huang et al. (2022) utilized the generalization of large language models to achieve zero-shot planners.

However, all these works may not generalize to a new distribution of dynamics or tasks since they encourage a strong connection between the given instruction and the policy, not the dynamics of the environment.

**Language grounding to dynamics of environments.** A different line of research has focused on utilizing manuals as auxiliary information to aid generalization. These text manuals provide

descriptions of the entities in the environment and their dynamics, *e.g.,* how they interact with other entities. Agents can figure out the dynamics of the environment based on the manual.

Narasimhan et al. (2018) explored transfer methods by simply concatenating the text description of an entity and the entity itself to facilitate policy generalization across tasks. RTFM (Zhong et al., 2019) builds the codependent representations of text manual and observation of the environment, denoted as $txt2\pi$, based on bidirectional feature-wise linear modulation (FiLM$^2$). A key challenge in RTFM is multi-modal multi-step reasoning with texts associated with multiple entities. EMMA (Hanjie et al., 2021) uses an entity-conditioned attention module that allows for selective focus over relevant descriptions in the text manual for each entity in the environment called MESSENGER. A key challenge in MESSENGER is the adversarial train-evaluation split without prior entity-text grounding. Recently, SILG (Zhong et al., 2021) unifies a collection of diverse grounded language learning environments under a common interface, including RTFM and MESSENGER. All these works have demonstrated the generalization of policies to a new environment with unseen entities and text descriptions.

Compared with the previous works, our work moves one step forward, investigating language grounding at the entity level in multi-agent settings.

**Subgoal Assignment.** In the goal-based multi-agent setting, in order to complete the goal more efficiently, agents need to coordinate with each other, *e.g.,* making a reasonable subgoal division. There are many language-free MARL models (Wang et al., 2020; 2019; Jeon et al., 2022; Tang et al., 2018; Yang et al., 2019) focusing on task allocation or behavioral diversity, which exhibit a similar effect as subgoal assignment.

However, without the help of inherited generalization of natural language, it is unlikely for the agent to perform well in environments unseen during training, which is supported by many previous works (Zhong et al., 2019; Hanjie et al., 2021). In addition, it is hard to depict the goal state sometimes without natural language (Liu et al., 2022), which makes the subgoal assignment even more challenging.

## 3 PRELIMINARIES

Our objective is to ground language to environment dynamics and entities for generalization to unseen multi-agent environments. Note that an entity is an object represented as a symbol in the observation, and dynamics refer to how entities behave in the environment.

**POSG.** We model the multi-agent decision-making problem as a partially observable stochastic game (POSG). For $n$ agents, at each timestep $t$, agent $i$ obtains its own partial observation $o_{i,t}$ from the global state $s_t$, takes action $a_{i,t}$ following its policy $\pi_\theta(a_{i,t}|o_{i,t})$, and receives a reward $r_{i,t}(s_t, \boldsymbol{a}_t)$ where $\boldsymbol{a}_t$ denotes the joint action of all agents. Then the environment transitions to the next state $s_{t+1}$ given the current state and joint action according to transition probability function $\mathcal{T}(s_{t+1}|s_t, \boldsymbol{a}_t)$. Agent $i$ aims to maximize the expected return $R_i = \sum_{t=1}^{T} \gamma^{t-1} r_{i,t}$, where $\gamma$ is the discount factor and $T$ is the episode time horizon.

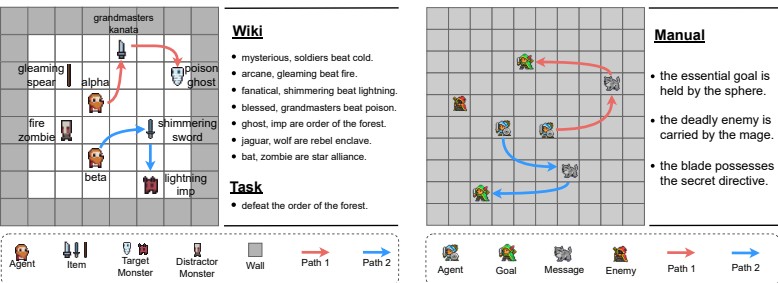

Figure 1: Illustrations of two modified multi-agent environments, *i.e.,* multi-agent RTFM (left) and multi-agent MESSENGER (right).

**Multi-Agent RTFM** is extended from Zhong et al. (2019), where for each task all agents are given the same text information based on collected human-written language templates, including a document of environment dynamics and an underspecified goal. Figure 1 illustrates an instance of the

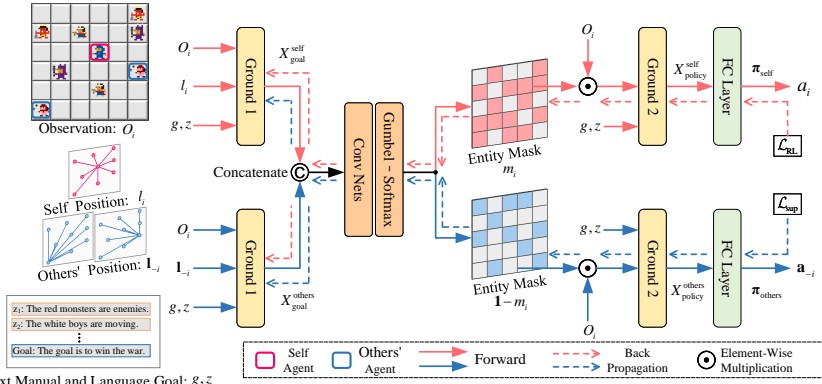

Figure 2: Overview architecture of EnDi. Language grounding is first used to obtain language-based representation $X_{goal}$. Then, subgoal division module generates a binary entity mask $m$ to address important entities. The mask is applied directly to the observation and new language-based representation $X_{policy}$ for decision-making is obtained. EnDi has two policy heads. One is to interact with the environment, and another is for agent modeling. The gradients from supervised learning and reinforcement learning jointly influence the formation of entity masks.

game. Concretely, the dynamics consist of monsters (*e.g.,* wolf, goblin), teams (*e.g.,* order of the forest), element types (*e.g.,* fire, poison), item modifiers (*e.g.,* fanatical, arcane), and items (*e.g.,* sword, hammer). When the agents encounter a weapon/monster, they can pick up the weapon/engage in combat with the monster. Moreover, a monster moves towards the nearest observable agent with $60\%$ probability, otherwise moves randomly. The general goal is to eliminate all monsters in a given team with the appropriate weapons. The game environment is rendered as a matrix of texts where each grid describes the entity occupying the grid.

**Multi-Agent MESSENGER** is built on Hanjie et al. (2021). For each task, all agents are provided with the same text manual. The manual contains descriptions of the entities, the dynamics, and the goal, obtained through crowdsourced human writers. In addition, each entity can take one of three roles: an enemy, a message, or a target. There are three possible movement types for each entity, *i.e.,* stationary, chasing, or fleeing. Agents are required to bring all the messages to the targets while avoiding enemies. If agents touch an enemy in the game or reach the target without first obtaining the message, they lose the game. Unlike RTFM, the game environment is rendered as a matrix of symbols without any prior mapping between the entity symbols and their descriptions.

In both multi-agent RTFM and multi-agent MESSENGER, each agent $i$ has an observation $o_t^i$ at each moment. By taking action $a_t^i$, each agent can get a reward $r_t^i$. The specific action space and reward rules are shown in Appendix A. Note that we choose these two environments because the effect of language grounding is well suited on them in single-agent scenarios. Also, the difference in manuals and rules of these two environments can help us explore the generalization ability of our EnDi method in the face of different language texts.

## 4 METHODOLOGY

In goal-based MARL, it is important for agents to coordinate with each other. Otherwise, subgoal conflicts (multiple agents focusing on the same subgoal) can impede the completion of the general goal. To this end, we introduce EnDi for language grounding in MARL to enable agents independently learn the subgoal division at the entity level for better generalization. For each task, agents are given a text manual $z \in Z$ describing the environment dynamics and a language-based goal $g \in G$ as language knowledge. Apart from this, at each timestep $t$, each agent $i$ obtains a $h \times w$ grid observation $o_{i,t}$ and outputs a distribution over the action $\pi(a_{i,t}|o_{i,t}, z, g)$. Note that we omit the subscript $t$ in the following for convenience. Note that parameters of each module are not shared among agents.

### 4.1 OVERALL FRAMEWORK

**Language Grounding Module.** Given a series of language knowledge, the agent first grounds the language to the observation of the environment, adopting the existing language grounding module

to generate a language-based representation $X = Ground(o, z, g) \in \mathbb{R}^{h \times w \times d,}$. It captures the relationship between the goal, the manual, and observation. Thus, agents are likely to understand the environment dynamics instead of memorizing any particular information, which is verified by previous works (Zhong et al., 2019; Hanjie et al., 2021). This type of representation is then used to generate the subgoal division and policy.

Note that EnDi is compatible with any language grounding module. In practice, we build our framework on the backbone of $txt2\pi$ (Zhong et al., 2019) and EMMA (Huang et al., 2022) and adopt their grounding modules, *i.e.,* FiLM$^2$ and multi-modal attention, respectively.

**Subgoal Division Module.** Since the language-based goal is highly relevant to the entities in the environment after grounding, the subgoal division should also be done at the entity level. To independently learn the good subgoal division, the goal is decomposed by each agent into two subgoals, the *self* subgoal it will focus on and the *others* subgoal which is exploited to avoid subgoal conflicts.

Taking agent $i$ to illustrate, it first generates two representations, *i.e.,* $X_{\text{goal}}^{\text{self}} = Ground1([o_i, l_i], z, g)$ and $X_{\text{goal}}^{\text{others}} = Ground1([o_i, \boldsymbol{l}_{-i}], z, g)$, where $l_i$ is the positional feature of agent $i$ which is the Manhattan distance to all the grids and $\boldsymbol{l}_{-i}$ is the joint positional feature of all agents except $i$. When grounding the general goal to entities $\boldsymbol{e} \in E$, different agents have different perspectives of how to achieve the goal, thus positional features are used to capture such differences. The two representations are then concatenated together and passed to a 2D convolution to extract a mixture feature map. With the mixture feature map, we utilize a Gumbel-Softmax (Jang et al., 2016) layer to output a subgoal division distribution over all the entities $\rho_{\boldsymbol{e}}(o_i, \boldsymbol{l}, z, g)$, determining what entities should be addressed.

To guarantee the uniform entities coverage of subgoal division module, we match the subgoal distribution $\rho_{\boldsymbol{e}}$ with a target distribution $p_{\boldsymbol{e}}^{\star}$ as a regularization term and aim to minimize $D_{\text{KL}}(\rho_{\boldsymbol{e}} || p_{\boldsymbol{e}}^{\star})$. As there is not any prior information for the subgoal division, we assume the target distribution has the mean of $|e|/n$ with a relatively low variance in the meantime, where $n$ is the number of agents. Intuitively, we are willing to observe that all agents make contributions to the goal. Any distribution with different mean or high variance might violate this motivation.

**Policy Module.** Agents try to understand the dynamics and the general goal through interactions with the entities. It means that, as the number of entities increases in MARL, the grounding procedure could be complex since more interactions are required during training.

Normally, unlike the general goal, agents only need to interact with parts of the relevant entities to complete the subgoal. Intuitively, if agents can ignore trivial entities, the grounding procedure can be simplified and accelerated at its source, and agents can make decisions without distractions.

To this end, we sample a binary entity mask $m_i$ based on the subgoal division distribution. The mask is applied to the observation directly and indicates the necessity of considering each entity when making decisions. Still, two representations are obtained as $X_{\text{policy}}^{\text{self}} = Ground2(o_i \odot m_i, z, g)$ and $X_{\text{policy}}^{\text{others}} = Ground2(o_i \odot (\mathbf{1} - m_i), z, g)$, where $\odot$ is the element-wise multiplication. There are two different policy heads in EnDi, $\pi_{\text{self}}$ and $\pi_{\text{others}}$. The former one is to output actions for interacting with the environment and takes $X_{\text{policy}}^{\text{self}}$ as input. On the other side, $X_{\text{policy}}^{\text{others}}$ is passed into another head for agent modeling, which will be elaborated on later. Note that $\pi_{\text{self}}$ is trained using reinforcement learning, and more training details can be found in Appendix B. In addition, Gumbel-softmax enables the gradients to flow through the binary entity mask to the subgoal division module.

## 4.2 AGENT MODELING

However, if EnDi does not consider the influence of others, the subgoal division module is prone to converge to a sub-optimum. In other words, agents may focus on individual interests rather than the common interest. Moreover, we want to capture the influence of others at the entity level for better coordination.

To this end, we let each agent reason about others' intentions by agent modeling. Specifically, EnDi has a policy head $\boldsymbol{\pi}_{\text{others}}$ based on $X_{\text{policy}}^{\text{others}}$ to predict the joint actions of others $\boldsymbol{a}_{-i}$ at each timestep

and update the policy parameter in a supervised manner,

$$\mathcal{L}(o_i, z, g) = -\boldsymbol{y} \log(\boldsymbol{\pi}_{others}(\boldsymbol{a}_{-i}|o_i, z, g))^\top, \tag{1}$$

where $\boldsymbol{y}$ is the concatenation of actual actions (one-hot vector) of others. By agent modeling, each agent can infer the entities that others plan to interact with since only accurate $\boldsymbol{1} - m_i$ can guarantee the low supervised loss. In other words, it would force the agent to avoid subgoal conflicts at the entity level.

## 5 EXPERIMENTS

We evaluate EnDi in multi-agent MESSENGER and multi-agent RTFM environments, which we have extended from two previous single-agent environments, see Section 3. For a fair comparison, we build EnDi based on EMMA and $txt2\pi$, which are the models in MESSENGER and RTFM, respectively. More details about the environments can be found in Appendix A.

### 5.1 CURRICULUM LEARNING

Language-based tasks are much more complex compared with language-free tasks since agents must go through a long process of exploration to learn to maximize the expected return and ground language in the meanwhile. To solve multi-agent MESSENGER and RTFM tasks (movable entities, complex game rules, multiple task goals), we design curricula to help policy learning. Note that millions of training steps are required for each training stage. It again demonstrates the complexity of such language-based environments, even dealing with grid worlds.

In **multi-agent MESSENGER**, we have designed Stages 1, 2, and 3, following settings of MESSENGER (Hanjie et al., 2021).

- **Stage 1 (S1)**: There are five entities (enemy, message1, message2, goal1, goal2) that are initialized randomly in five possible locations. Agents always begin in the center of the grid. They start without messages with a probability 0.8 and begin with messages otherwise. On S1, agents are required to obtain different messages (start without messages)/goals (start with messages) to win. If all agents interact with the correct entities within a given number of steps, a reward of $+1$ will be provided, and $-1$ otherwise.

- **Stage 2 (S2)**: The five entities are initialized randomly in five possible locations. The difference with S1 is that the entities on S2 are mobile. In other words, they can change position randomly at every timestep.

- **Stage 3 (S3)**: The five entities are initialized randomly in five possible locations. All agents begin without the message, and the rule requires them to interact with the messages first and then with the goals. Within a given number of steps, each agent interacting successfully with the messages will be provided a reward of 0.2, and all agents interacting successfully with the goals will obtain a reward of $+1$, otherwise $-1$.

In **multi-agent RTFM**, there are five stages for training. Note that Stage 2 to Stage 5 follow settings of RTFM (Zhong et al., 2019), while we empirically found Stage 1 is beneficial for later training process in multi-agent setting. During every episode, we subsample a set of groups, monsters, modifiers, and elements to use. We randomly generate group assignments of which monsters belong to which team and which modifier is effective against which element. Note that each assignment represents a different dynamic.

- **Stage 1 (S1)**: There are only four entities in the environment, *i.e.,* two correct items and two monsters. Note that there are one-to-one group assignments, stationary monsters, and no templated language descriptions. Agents need to pick up the correct items and use them to beat all the monsters to win within a given number of steps.

- **Stage 2 (S2)**: We add two distractor entities (sampled from the group that is irrelevant to the goal). One is a distractor weapon that cannot be used to beat the monsters. Another is a distractor monster, with which agents will lose when engaging.

- **Stage 3 (S3)**: All monsters, including the distractor monsters, move towards the agents with $60\%$ probability, otherwise move randomly.

- **Stage 4 (S4)**: We allow many-to-one group assignments to make disambiguation more difficult. Note that the difference between many-to-one and one-to-one groups is that the manual would also describe the entities that are not presented in the game.

- **Stage 5 (S5)**: The manual would use templated language descriptions.

Table 1: The final mean win rate ($\pm$ stddev.) over five random seeds. All methods adopt the same curriculum training strategy for a fair comparison. Note that multi-agent RTFM has five training stages, while multi-agent MESSENGER only needs three stages. $txt2\pi$, EMMA, and SILG are the single-agent baselines of previous works. Removing the supervised loss from the total loss, denoted as EnDi-sup, and removing the regularization term, denoted as EnDi-reg. EnDi(num) and EnDi(dis) are our methods with different form of regularization terms.

| | Model | S1 | S2 | S3 | S4 | S5 |
|---|---|---|---|---|---|---|
| multi-agent RTFM | $txt2\pi$ | $0.99 \pm 0.01$ | $0.11 \pm 0.06$ | $0.05 \pm 0.05$ | $0.05 \pm 0.03$ | $0.02 \pm 0.01$ |
| | SILG | $0.99 \pm 0.01$ | $0.05 \pm 0.01$ | $0.04 \pm 0.01$ | $0.04 \pm 0.00$ | $0.04 \pm 0.01$ |
| | EnDi-sup | $0.99 \pm 0.01$ | $0.53 \pm 0.08$ | $0.36 \pm 0.03$ | $0.41 \pm 0.12$ | $0.41 \pm 0.13$ |
| | EnDi-reg | $1.00 \pm 0.00$ | $0.61 \pm 0.15$ | $0.39 \pm 0.01$ | $0.39 \pm 0.03$ | $0.39 \pm 0.03$ |
| | EnDi(num) | $1.00 \pm 0.00$ | $0.62 \pm 0.07$ | $0.52 \pm 0.05$ | $0.54 \pm 0.03$ | $0.56 \pm 0.01$ |
| | EnDi(dis) | $1.00 \pm 0.00$ | $\mathbf{0.76 \pm 0.14}$ | $\mathbf{0.59 \pm 0.04}$ | $\mathbf{0.61 \pm 0.04}$ | $\mathbf{0.63 \pm 0.03}$ |
| multi-agent MESSENGER | EMMA | $0.57 \pm 0.03$ | $0.04 \pm 0.02$ | $0.02 \pm 0.01$ | — | — |
| | SILG | $0.99 \pm 0.01$ | $0.32 \pm 0.04$ | $0.05 \pm 0.02$ | — | — |
| | EnDi-sup | $0.99 \pm 0.01$ | $0.52 \pm 0.05$ | $0.21 \pm 0.03$ | — | — |
| | EnDi-reg | $0.67 \pm 0.03$ | $0.07 \pm 0.02$ | $0.02 \pm 0.01$ | — | — |
| | EnDi(num) | $0.99 \pm 0.01$ | $0.75 \pm 0.04$ | $0.19 \pm 0.02$ | — | — |
| | EnDi(dis) | $0.99 \pm 0.01$ | $\mathbf{0.81 \pm 0.03}$ | $\mathbf{0.25 \pm 0.03}$ | — | — |

## 5.2 RESULTS

In the experiments, we compare EnDi with the following methods: 1) $txt2\pi$ (Zhong et al., 2019) which builds the codependent representations of text manual and observation of the environment based on bidirectional feature-wise linear modulation ($\text{FiLM}^2$) in RTFM. 2) EMMA (Hanjie et al., 2021) which uses an entity-conditioned attention module that allows for selective focus over relevant descriptions in the text manual for each entity in MESSENGER. 3) SILG (Zhong et al., 2021) that provides the first shared model architecture for several symbolic interactive environments, including RTFM and MESSENGER. For all methods, we train an independent model for each agent separately.

**Performance.** Table 1 shows the win rates in both multi-agent RTFM and multi-agent MESSENGER environments. In each environment, there are two agents. In the multi-agent RTFM environment, we use EnDi with $txt2\pi$ as the backbone to compare with both $txt2\pi$ and SILG baselines. It is worth noting that as discussed in Section 4.1, to guarantee the uniform entities coverage of subgoal division module, we match the subgoal distribution $\rho_e$ with a target distribution $p_e^\star$ as a regularization term. Specifically, we design two patterns for the regularization term, 1) EnDi(num) which minimizes $||\boldsymbol{e}_i| - |\boldsymbol{e}_{-i}|/(n-1)|$ where $|\boldsymbol{e}_i|$ is the entity number of the entity mask for agent $i$, and 2) EnDi(dis) which minimizes the distance to reach the divided entities, *i.e.,* $\sum_{k=\{i,-i\}} \sum_{e \in \boldsymbol{e}_k} (|x_e - x_k| + |x_e - y_k|)$ where $(x, y)$ denotes the coordinate of entity/agent in the mask. In multi-agent RTFM, all models are able to succeed in the simplest S1, achieving a 99% to 100% win rate. When gradually transferring to S2–S5, both EnDi(dis) and EnDi(num) consistently show significantly better win rates than baselines.

We also conduct similar experiments in multi-agent MESSENGER, where we use EnDi that builds on EMMA to compare with both EMMA and SILG baselines. EnDi and SILG both succeed in S1 while EMMA obtains a win rate of 57%. Then, EMMA is only able to achieve a 4% win rate in S2, while EnDi can obtain a win rate at most 81%, much better than SILG and EMMA. In S3, the environment is dynamically changing and the tasks are complex, making it extremely difficult to win. Even in the single-agent version, the best result is only a 22% win rate (Hanjie et al., 2021). In the multi-agent version, the complex agent-entity interactions make S3 even more difficult. All models exhibit limited win rates due to the complexity of the environment and tasks, but EnDi(num) and EnDi(dis) still show a substantial win rate lead over baselines.

The consistent results in both environments highlight the importance and rationality of EnDi's design of subgoal division, demonstrating its effective performance in multi-agent settings. In addition, the poor performance of single-agent methods indicates that without coordination behaviors at the entity level, they cannot handle more challenging multi-agent settings.

**Policy Analysis.** To verify the effect of EnDi on the subgoal division, we visualize the learned policies using EnDi. Figure 3 shows key snapshots from the learned policies on randomly sampled multi-agent RTFM and multi-agent MESSENGER tasks.

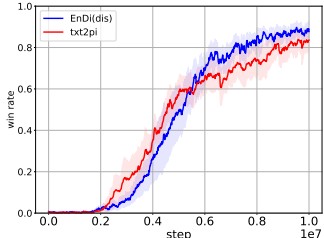

Figure 3: Illustration of learned subgoal division in multi-agent RTFM (*upper panel*) and multi-agent MESSENGER (*lower panel*). The dash circle shows the chosen subgoal for each agent.

The upper panel in Figure 3 shows an episode from a multi-agent RTFM task. In this task, EnDi can reasonably factor teamwork into the proper subgoal division and complete the task quickly rather than simply finishing the subgoal nearby. In frames 1–3, although Agent1 has a closer weapon in the upper left corner, it ignores this weapon and chooses another weapon at the top of the map. This decision helps Agent2 to differentiate from Agent1 in its subgoal division, and also avoids the distractor monster on the left side of the map. In frames 3–5, it can be seen that after Agent2 gets the weapon, it chooses to go down to the target monster, leaving the upper target monster to Agent1 and thus forming a coordinated strategy between the two agents.

The lower panel in Figure 3 shows an episode from multi-agent MESSENGER. We can observe similar subgoal division strategies as multi-agent RTFM. In more detail, the message at the upper left is closer to Agent1 than the lower left one in frame 1. However, to avoid subgoal conflicts, Agent1 chooses another subgoal, which is the message at the lower left. For the rest frames, two agents go to complete their chosen subgoals so the general goal can be finished more efficiently.

**Ablations.** Compared to $txt2\pi$ or EMMA, EnDi has two additional modules: the subgoal division module and the agent modeling module. To investigate the effectiveness of each module, we compare EnDi(dis) with two ablation baselines. We remove the supervised loss from the total loss, denoted as EnDi-sup, and remove the regularization term, denoted as EnDi-reg. The results are shown in Table 1. Note that the model degrades to $txt2\pi$ or EMMA if we remove both modules.

As shown in Table 1, EnDi(dis) achieves a higher win rate than EnDi-sup and EnDi-reg in both environments. These results verify the importance of considering the influence of others and regularizing the subgoal division distribution. In more detail, we found that no subgoal division strategy can be learned in EnDi-reg, where the entity mask converges to a poor sub-optimum, *i.e.,* covering almost all the entities. As for EnDi-sup, we observed that conflicts of the chosen subgoals between agents happen more often than the model with agent modeling.

**Extension.** EnDi can also adapt to environments with more than two agents. Specifically, the joint positional feature of all other agents is the Manhattan distance of all the grids to their closest agents. For agent modeling, we have multiple policy heads for all other agents.

To verify this, we consider three-agent RTFM and set the number of items and monsters to three. Figure 4 shows the learning curves of EnDi(dis) and $txt2\pi$. The result is presented in terms of the mean and standard deviation of five runs with different random seeds. Although the S1 of multi-agent RTFM is easy for all the baselines, EnDi(dis) can still get a higher win rate. This demonstrates that EnDi(dis) can adapt to more complex settings, *i.e.,* more agents. More thorough investigation is left as future work.

Figure 4: Learning curve of in terms of the win rate in S1 of three-agent RTFM.

## 5.3 GENERALIZATION

As mentioned before, the biggest advantage of language-based MARL is that natural language drives the generalization of policies. To investigate the generalization ability of EnDi, we zero-shot transfer the model learned from the training tasks to new tasks. Note that we follow the evaluation rules in RTFM (Zhong et al., 2019) and MESSENGER (Hanjie et al., 2021).

To assess the importance of language, EMMA (Hanjie et al., 2021) has done the ablation studies by replacing the language descriptions with an auxiliary vector (Game ID) where each dimension corresponds to a role. The results show the Game ID fails in the complex environment. We also test the generalization of language-free method, QMIX (Rashid et al., 2018). At the simplest S1 in multi-agent RTFM, the mean win rates of QMIX are *only* $64\%$ and $41\%$ in the training set and unseen tasks, respectively, much worse than EnDi. These results distinguish EnDi from language-free methods.

**Multi-Agent RTFM.** Importantly, no assignments of monster-team-modifier-element are shared between training and evaluation set (denoted as Eval). Thus, even with the same words, the new assignments still lead to new dynamics. However, we also design another evaluation set with the totally novel monsters and modifiers, denoted as Eval(new). Moreover, since we train our model in the $8 \times 8$ grid world, we also test the generalization to a bigger $10 \times 10$ grid world in the original evaluation set, denoted as Eval($10 \times 10$). Note that all results are obtained by models from S5.

Table 2: The mean win rates ($\pm$ stddev.) over five seeds on evaluation in multi-agent RTFM. Note that the result of each seed is obtained by running 1k episodes.

|  | S5-Train | S5-Eval | S5-Eval(new) | S5-Eval($10 \times 10$) |
|---|---|---|---|---|
| EnDi-sup | $0.38 \pm 0.11$ | $0.39 \pm 0.10$ | $0.36 \pm 0.10$ | $0.44 \pm 0.11$ |
| EnDi-reg | $0.39 \pm 0.02$ | $0.38 \pm 0.02$ | $0.35 \pm 0.02$ | $0.41 \pm 0.00$ |
| EnDi(dis) | $0.59 \pm 0.01$ | $0.61 \pm 0.01$ | $0.56 \pm 0.00$ | $0.62 \pm 0.02$ |

Table 2 shows that EnDi(dis) and its variants have demonstrated an extremely promising generalization ability to new dynamics unseen during training or a different size of grid world. This benefits from grounding language to the dynamics. In addition, we notice that EnDi-reg performs slightly worse in Eval(new) compared with others. We speculate the language grounding process may be slightly hard in EnDi-reg since the entity mask of EnDi-reg covers more entities than others.

**Multi-Agent MESSENGER.** For evaluation, we still keep the same entity types, but we make new assignments of word combinations for message, goal, and enemy to ensure that these combinations are unseen during training. We also use completely new description sentences in the text manuals.

The results are shown in Table 3. EnDi(dis) shows the promising generalization ability in multi-agent MESSENGER. EnDi(dis) can maintain 79% and 52% win rates on S1 and S2 respectively, playing with completely unseen entity combinations and text manuals. Due to the difficulty of the S3, all models underperform but EnDi(dis) still maintains the best generalization performance. In addition, the problem of EnDi-reg to handle language grounding is more magnified in multi-agent MESSENGER, yielding a lower win rate in both training and evaluation. This result demonstrates the importance of focusing on relevant entities when making decisions.

Table 3: The mean win rates ($\pm$ stddev.) over five seeds on evaluation in multi-agent MESSENGER. Note that the result of each seed is obtained by running 1k episodes.

|  | S1-Train | S1-Eval | S2-Train | S2-Eval | S3-Train | S3-Eval |
|---|---|---|---|---|---|---|
| EnDi-sup | $0.99 \pm 0.01$ | $0.68 \pm 0.07$ | $0.52 \pm 0.05$ | $0.36 \pm 0.06$ | $0.21 \pm 0.03$ | $0.08 \pm 0.03$ |
| EnDi-reg | $0.67 \pm 0.03$ | $0.39 \pm 0.04$ | $0.07 \pm 0.02$ | $0.04 \pm 0.01$ | $0.02 \pm 0.01$ | $0.01 \pm 0.01$ |
| EnDi(dis) | $0.99 \pm 0.01$ | $0.79 \pm 0.05$ | $0.81 \pm 0.03$ | $0.52 \pm 0.02$ | $0.25 \pm 0.03$ | $0.11 \pm 0.02$ |

## 6 CONCLUSION

We attempt to address the challenges of grounding language for generalization to unseen dynamics in multi-agent settings. To this end, we have proposed a novel language-based framework, EnDi, that enables agents independently learn subgoal division strategies at the entity level. Empirically, EnDi outperforms existing language-based methods in all tasks by a large margin and demonstrates a promising generalization ability. Moreover, we conclude that many language grounding problems can be addressed at the entity level.

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

# A ENVIRONMENT DETAILS

## A.1 MULTI-AGENT RTFM

In multi-agent RTFM, the agents are given a document of environment dynamics, observations of the environment, and an underspecified goal instruction. Concretely, we design a set of dynamics that consists of monsters *e.g.,* wolf or goblin, teams, *e.g.,* Order of the Forest, element types, *e.g.,* fire or poison, item modifiers, *e.g.,* fanatical or arcane, and items, *e.g.,* sword or hammer. When the player is in the same grid with a monster or weapon, the player picks up the item or engages in combat with the monster. The player can possess one item at a time, and drops existing weapons if they pick up a new weapon. A monster moves towards the nearest observable agent with $60\%$ probability, otherwise moves randomly. The dynamics, the agents' inventories, and the underspecified goal are rendered as text. The game world is rendered as a matrix of texts in which each grid describes the entity occupying the grid. We use human-written templates for stating which monsters belong to which team, which modifiers are effective against which element, and which team the agent should defeat.

For each episode, multi-agent RTFM subsamples a set of groups, monsters, modifiers, and elements to use. Multi-agent RTFM randomly generates group assignments of which monsters belong to which team and which modifier is effective against which element. A document that consists of randomly ordered statements corresponding to this group assignment is presented to the agents. Multi-agent RTFM samples one element, one team, and a monster from that team, *e.g.,* "fire goblin" from "Order of the forest", to be the target monster. Additionally, we sample one modifier that beats the element and an item to be the item that defeats the target monster, *e.g.,* "fanatical sword". Similarly, we sample an element, a team, and a monster from a different team to be the distractor monster, *e.g.,* poison bat, as well as an item that defeats the distractor monster, *e.g.,* arcane hammer.

In the multi-agent setting, we set the number of correct items and the number of target monsters to be the same as the number of agents. The number of the distractor item and the distractor monster is set to one. Under this setting, to achieve the goal more efficiently, the agents need to coordinate with each other. One agent can solve the game alone, but need more steps.

**Game Rules.** Once agents kill all the correct monsters, they win the game. Any agent who kills a correct monster can get a reward $+1$.

If any agent engages with the distractor monster, they lose the game and whoever engages gets a reward $-1$. Moreover, if they exceed the steps given by the game or they kill the distractor monster with the distractor item, they still lose the game. For each step, all agents get a penalty of $-0.02$. Note that the reward is not shared across all agents and this is different with multi-agent MESSENGER.

In order to win the game, each agent needs to do the following things:

- Identify the target team from the goal.
- Identify the monsters that belong to that team.
- Identify which monster is in the world.
- Identify the modifiers that are effective against this element.
- Find which modifier is present and the item with the modifier.
- Figure out what item should be picked up among all the correct items in order to avoid miscoordination.
- Pick up the correct item.
- Figure out what monsters should be engaged among all the correct items in order to avoid miscoordination.
- Engage the correct monster in combat.

**Entities and Modifiers.** Below is a list of entities and modifiers contained in multi-agent RTFM:

- Monsters: wolf, jaguar, panther, goblin, bat, imp, shaman, ghost, zombie.
- Weapons: sword, axe, morningstar, polearm, knife, katana, cutlass, spear

- Elements: cold, fire, lightning, poison.
- Modifiers: Grandmaster, blessed, shimmering, gleaming, fanatical, mysterious, soldier, arcane.
- Teams: star alliance, order of the forest, rebel enclave.

For the Eval-new set, we have a new list of entities and modifiers:

- Monsters: tiger, bear, puma, elf, vampire, gremlin, witch, specter, robot.
- Weapons: sabre, tomahawk, sunglow.
- Modifiers: superstars, sacred, glittering, shiny, obsessive, bizarre, secret, esoteric.

**Language Templates.** Multi-agent RTFM collects human-written language templates for the goal and the dynamics (Zhong et al., 2019). The goal statements in multi-agent RTFM describe which team the agent should defeat. Multi-agent RTFM collects 12 language templates for goal statements. The document of environment dynamics consists of two types of statements. The first type describes which monsters are assigned to which team. The second type describes which modifiers, which describe items, are effective against which element types, which are associated with monsters. Multi-agent RTFM collects 10 language templates for each type of statement. The entire document is composed of statements, which are randomly shuffled. Multi-agent RTFM randomly samples a template for each statement, which multi-agent RTFM fills with the monsters and team for the first type and modifiers and elements for the second type.

## A.2 MULTI-AGENT MESSENGER

Multi-Agent MESSENGER is built on Hanjie et al. (2021). In order to transform single-agent MES-SENGER into multi-agent MESSENGER, we keep the same entity, role, and adjective as used in single-agent MESSENGER, and increase only the number of entities. For each task, all agents are provided with the same text manual. The manual contains descriptions of the entities, the dynamics, and the goal, obtained through crowdsourced human writers. Crucially, while prior work assumes a ground truth mapping (*e.g.,* the word 'queen' in the manual refers to the entity name 'queen' in the observation), multi-agent MESSENGER does not contain priors that map between text and state observations. To succeed in multi-agent MESSENGER, an agent must relate entities and dynamics of the environment to their references in the natural language manual using only scalar reward signals from the environment. The overall game mechanics of multi-agent MESSENGER involve obtaining a message and delivering it to a goal.

In multi-agent MESSENGER, each entity can take one of three roles: an enemy, a message, or a target. There are three possible movement types for each entity, *i.e.,* stationary, chasing, or fleeing. Agents are required to bring all the messages to the targets while avoiding enemies. If agents touch an enemy in the game or reach the target without first obtaining the message, they lose the game. There are twelve different entities. Each set of entity-role assignments is initialized on a $10 \times 10$ grid. The agent can navigate via up, down, left, right, and stay actions and interacts with another entity when both occupy the same grid. The same set of entities with the same movements may be assigned different roles in different games. Thus, two games may have identical observations but differ in the reward function (which is not available to the agents) and the text manual (which is available). Thus, the agents must learn to extract information from the text manual to succeed consistently.

**Game Rules.** In order to win the game, each agent needs to do the following things,

- Identify the entities that hold the message.
- Map descriptions to the correct symbols in the observation.
- Observe the movement patterns of entities to disambiguate which of the two entities holds the message.
- Pick up the messages from the entities that hold them.
- Follow a similar procedure to the third step to disambiguate which mages are the goals and which are the enemies.

- Bring the messages to the goals.

In different stages, the reward functions also have the following differences:

- S1 & S2: If all the agents interact with the correct entities (*i.e.,* agents started with message obtain a goal and agents started without message obtain a message) within a given number of steps, a reward of $1.0$ will be provided, and $-1.0$ otherwise.
- S3: The agents are required to get the messages first and then the goals so that it can be considered a win. Within a given number of steps, each agent interacting successfully with the messages will be provided a reward of $0.2$, and all the agents interacting successfully with the goals will receive a reward of $1.0$, otherwise $-1.0$.

**Entities.** Below is a list of entities contained in multi-agent MESSENGER:

- Enemy: enemy, opponent, adversary (Adjectives: dangerous, deadly, lethal).
- Message: message, memo, report. (Adjectives: restricted, classified, secret)
- Goal: goal, target, aim. (Adjectives: crucial, vital, essential)

**Language Templates.** We use the same corpus of text manual as in the original EMMA paper (Hanjie et al., 2021), *i.e.,* 82 templates with 2214 possible descriptions after filling in the blanks. We have three blanks per template, one each for the entity, role, and adjective. For each role, we have three role words and three adjectives that are synonymous. Each entity is also described in three synonymous ways. Thus, every entity-role assignment can be described in 27 different ways on the same template. The raw templates are filtered for duplicates, converted to lowercase, and corrected for typos to prevent confusion in specific tasks.

## B  IMPLEMENTATION AND TRAINING DETAILS

First, for both environments, we train 5 runs with different random seeds for each baseline to report the mean and standard deviation. For generalization, we zero-shot transfer each of the 5 runs to the new task and once again report the mean and standard deviation.

### B.1  MULTI-AGENT RTFM

We train using an implementation of IMPALA (Espeholt et al., 2018). In particular, we use 10 actors and a batch size of 20. When unrolling actors, we use a maximum unroll length of 80 steps. Each episode lasts for a maximum of 1000 steps. We optimize using RMSProp with a learning rate of 0.005, which is annealed linearly for 100 million steps. We set $\alpha = 0.99$ and $\epsilon = 0.01$.

During training, we apply a small negative reward for each time step of $-0.02$ and a discount factor of $0.99$ to facilitate convergence. We additionally include an entropy regularization to encourage exploration. RTFM sets in the entropy loss with a weight of 0.005 and the baseline loss with a weight of 0.5. The baseline loss is computed as the root mean square of the advantages (Espeholt et al., 2018).

For S1, we train the model for $1.5 \times 10^7$ steps. For the rest stages, *i.e.,* S2–S5, we train the model for $5 \times 10^7$ steps.

### B.2  MULTI-AGENT MESSENGER

The models are end-to-end differentiable and we train them using proximal policy optimization (PPO) (Schulman et al., 2017) with $\gamma = 0.99$, $\epsilon = 0.01$ and the Adam optimizer with learning rate $\alpha = 5 \times 10^5$. On S1, S2, and S3, we limit each episode to 4, 32, and 64 steps respectively. We train models for up to 12 hours on S1, 72 hours on S2, and 144 hours on S3. We use the validation games to save the model parameters with the highest validation win rate during training and use these parameters to evaluate the models on the test games.

For S1, we train the models for $3 \times 10^7$ steps. For S2 and S3, we train the model for $5 \times 10^7$ steps.

# C    MODEL DESIGN

We have two versions of EnDi. One takes the $txt2\pi$ as backbone. Another takes EMMA as backbone.

## C.1    $txt2\pi$ VERSION

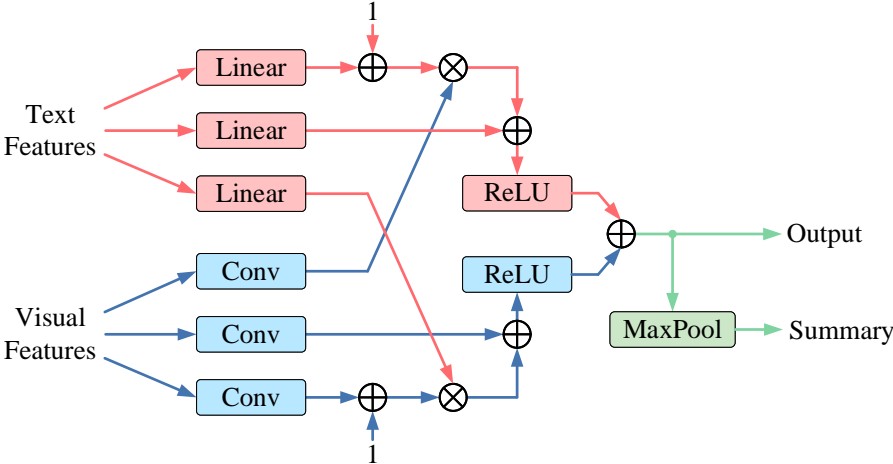

Figure 5: Architecture of FiLM². FiLM² builds codependent representations of text and visual inputs by modulating each kind of input using representations of input in another modality.

For the language grounding module, we utilize FiLM² to capture the relationship between language knowledge and environment observations. The architecture of FiLM² is shown in Figure 5. Not that each FiLM² layer uses $3 \times 3$ convolutions and padding and stride sizes of 1. In more detail, the grounding module that is used to generate $X_{\text{goal}}^{\text{self}}$ and $X_{\text{goal}}^{\text{others}}$ is shared and consists of two FiLM² layers with channel of 16 and 4. The convolution layer that processes the concatenation of $X_{\text{goal}}^{\text{self}}$ and $X_{\text{goal}}^{\text{others}}$ uses kernel of $3 \times 3$ and outputs 2 dimension features. Moreover, the grounding module that is used to generate $X_{\text{policy}}^{\text{self}}$ and $X_{\text{policy}}^{\text{others}}$ are shared and consists of one FiLM² layer with channel of 16, 32, 64, 64, and 64, with residual connections from the 3rd layer to the 5th layer. The two representations need to get a polling operation over spatial dimension before going through the corresponding policy head (one fully-connected layer).

For other text pre-processing modules, we strictly follow the RTFM setting (Zhong et al., 2019). The Bidirectional LSTM (BiLSTM) that processes the inventory and the goal has a hidden dimension of size 10. The BiLSTM that processes the document has a hidden dimension of size 100. Note that we use a word embedding dimension of 30.

For more details about the $txt2\pi$, please refer to the paper (Zhong et al., 2019).

## C.2    EMMA VERSION

We follow the EMMA settings to both capture the relationship between the language knowledge and the environment observations, and pre-process the language manual in the MESSENGER environment. The architecture of EMMA is shown in Figure 6. The EMMA model consists of 3 components including Text Encoder, Entity Representation Generator and Action Module. In Text Encoder, the input consists of a $h \times w$ grid observation with a set of entity descriptions. EMMA encodes each description using a **BERT-base model** whose parameters are fixed throughout training. Then the key and value vectors are obtained from the encoder. In Entity Representation Generator, EMMA embeds each entity's symbol into a query vector to attend to the descriptions with their respective key and value vectors. For each entity $e$ in the observation, EMMA places its representation $x_e$ into a tensor $X \in \mathbb{R}^{h \times w \times d}$ at the same coordinates as the entity position in the observation to maintain full spatial information. The representation for the agent is simply a learned embedding of dimension

*d.* In Action Module, to provide temporal information that assists with grounding movement dynamics, EMMA concatenates the outputs of the representation generator from the three most recent observations to obtain a tensor $X' \in \mathbb{R}^{h \times w \times 3d}$. To get a distribution over the actions, EMMA runs a 2D convolution on $X'$ over the $h, w$ dimensions. The flattened feature maps are passed through a fully-connected FFN terminating in a softmax over the possible actions.

For more details about EMMA, please refer to the paper (Hanjie et al., 2021).

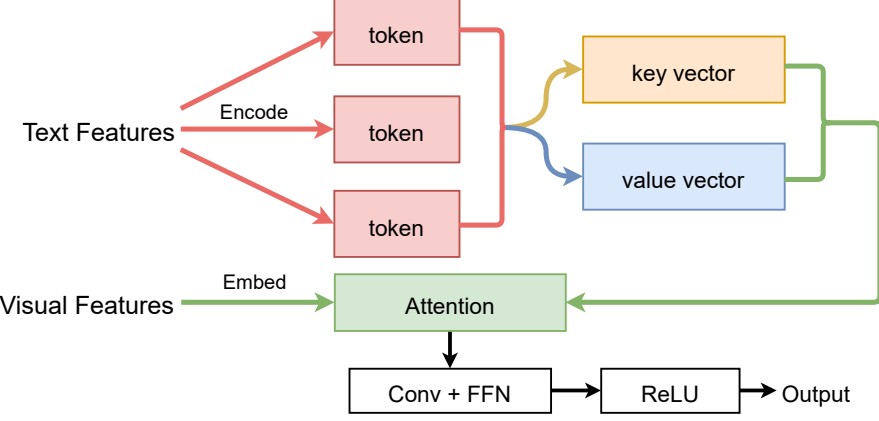

Figure 6: Architecture of EMMA.

## D    ADDITIONAL RESULTS

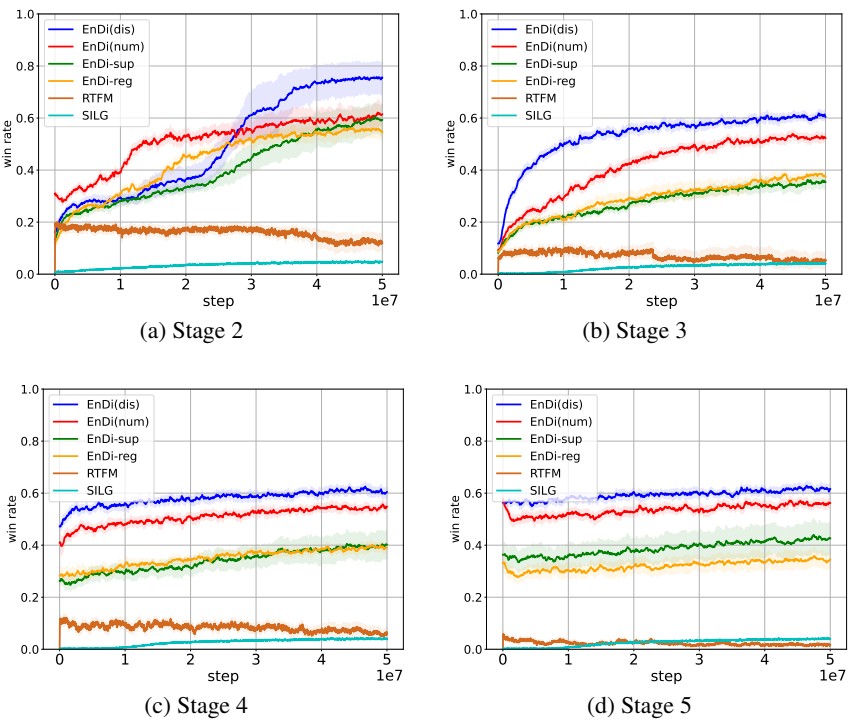

(a) Stage 2

(b) Stage 3

(c) Stage 4

(d) Stage 5

Figure 7: Learning curves in terms of the win rate of EnDi and baselines in multi-agent RTFM.

In this section, we provide the learning curve of EnDi and baselines. All results are presented in terms of the mean and standard deviation of five runs with different random seeds.

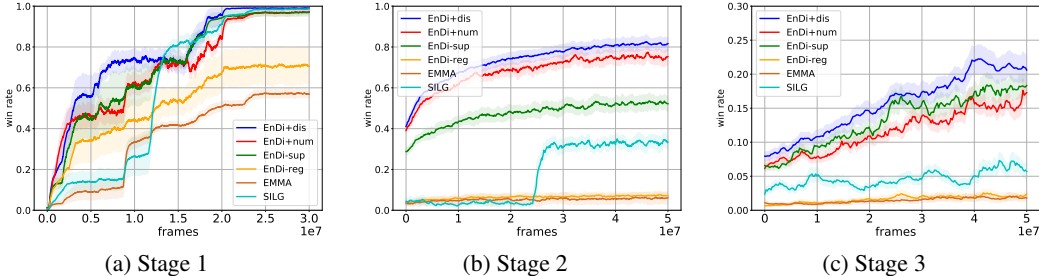

(a) Stage 1          (b) Stage 2          (c) Stage 3

Figure 8: Learning curves in terms of the win rate of EnDi and baselines in multi-agent MESSEN-GER.

Figure 7 shows the results in multi-agent RTFM from Stage 2 to Stage 5. Note that Stage 1 is omitted since all baselines achieve nearly 100% win rate. Figure 8 shows the results in multi-agent MESSENGER from Stage 1 to Stage 3. The results again demonstrate the superiority of our method over baselines.

# E  RESULTS WITHOUT VALIDATION

Table 4: The mean win rates ($\pm$ stddev.) of EnDi(dis) without validation over five seeds on training in multi-agent MESSENGER. Note that the result of each seed is obtained by running 1k episodes.

| EnDi(dis) - train | w/ validation | w/o validation |
|---|---|---|
| S1 | $0.99 \pm 0.01$ | $0.99 \pm 0.01$ |
| S2 | $0.81 \pm 0.03$ | $0.76 \pm 0.02$ |
| S3 | $0.25 \pm 0.03$ | $0.20 \pm 0.04$ |

Table 5: The mean win rates ($\pm$ stddev.) of EnDi(dis) without validation over five seeds on evaluation in multi-agent MESSENGER. Note that the result of each seed is obtained by running 1k episodes.

| EnDi(dis) - test | w/ validation | w/o validation |
|---|---|---|
| S1 | $0.79 \pm 0.05$ | $0.77 \pm 0.02$ |
| S2 | $0.52 \pm 0.02$ | $0.48 \pm 0.03$ |
| S3 | $0.11 \pm 0.02$ | $0.09 \pm 0.02$ |

In order to make a fair comparison, we followed the original EMMA setting in the paper and added the validation procedure. But to verify the performance of EnDi more strictly, we conduct additional ablation experiments without the validation procedure. As shown in Table 4 and 5, the performance of EnDi is only weakly affected by the removal of validation and still outperforms the baselines. Therefore, validation does not affect the conclusions in the experimental results of the paper.

# F  ERROR ANALYSIS

## F.1  ERROR CASE IN MULTI-AGENT RTFM

**Case 1** (Figure 9, *upper panel*): the entity division is reasonable in Multi-agent RTFM: Fire Imp (target monster), Shimmering Sword (good weapon) and Poison Zombie (distractor monster) belongs to the entity division of agent 1, while Fanatical Kanata (good weapon), Lightning Ghost (target

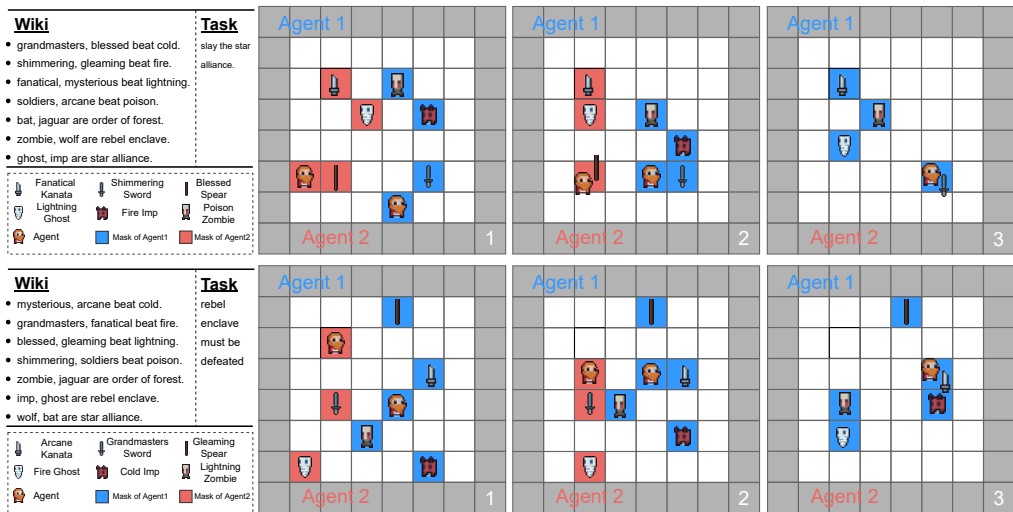

Figure 9: Error cases of EnDi in Multi-agent RTFM.

monster) and Blessed Spear (bad weapon) belongs to the entity division of agent 2. Each agent's subgoal includes a good weapon and a target monster. However, agent 2 picks up the bad weapon (Blessed Spear), resulting in it being killed by the monster (Lighting Ghost) that chased it on the way to the good weapon(Fanatical Kanata), although Lighting Ghost is the target monster.

**Case 2** (Figure 9, *lower panel*): the entity division is also rational: Cold Imp (target monster), Arcane Kanata (good weapon), Lightning Zombie (distractor monster) and Gleaming Spear(bad weapon) belongs to the entity division of agent 1, while Fire Ghost(target monster) and Grandmasters Sword (good weapon) belongs to the entity division of agent 2. Agent 2 choose to pick up the good weapon (Grandmasters Sword), but at that point it's beat by the distractor monster (Lightning Zombie) that came after it.

All in all, we can blame the error case to that agents recognize the wrong entities. In other words, the language grounding procedure may suffer from the multi-agent settings.

## F.2    ERROR CASE IN MULTI-AGENT MESSENGER

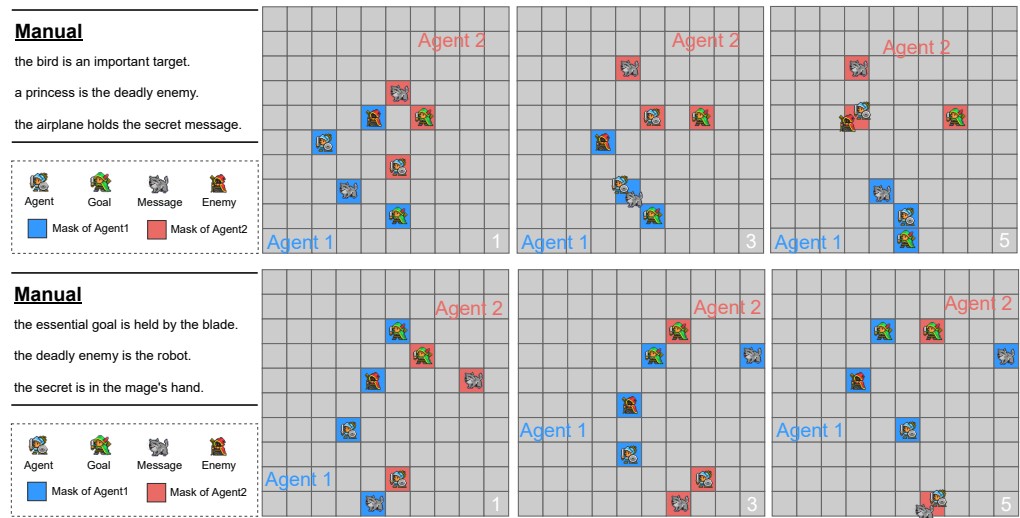

Figure 10: Error cases of EnDi in Multi-agent MESSENGER.

In frame 1 of Case 1 (Figure 10, *upper panel*), the entity divider reasonably assigns the entities to both agents. But in frame 3, the enemy is close to both agents but the entity divider can only assign the enemy to agent 1, resulting in agent 2 not paying good attention to the enemy and being killed by the enemy in frame 5. In the future, we consider to solve this problem by adding some sharing elements to the entity divider.

Another unexpected situation is shown in Case 2 (Figure 10, *lower panel*), where both agents are far away from the message on the right side in frame 1, and the entity divider can only assign it to one agent. But in frame 3, due to the movements of the message and the goal, the entity divider chooses to reassign the message, causing agent 1 to change its direction from downward to upward after frame 3. Although the agents finally complete the task, the invalid movement in the first few frames makes the agents less efficient. We will consider a more reasonable assignment way to improve the entity divider.

