# OpenReview forum: "Entity Divider with Language Grounding in Multi-Agent Reinforcement Learning"
_ICLR.cc/2023/Conference — Submitted to ICLR 2023_

### Official Review · Reviewer_imbM · 2022-10-24

**Confidence:** 3
**Correctness:** 3
**Technical Novelty And Significance:** 3
**Empirical Novelty And Significance:** 3
**Recommendation:** 6

**Clarity, Quality, Novelty And Reproducibility:**

Quality:

Experiments are mostly thorough. I would have liked to see a bit of error analysis to see what kind of mistakes are still being made in this environment.

Clarity:

I struggled to understand the task setting in a formal way.
- Throughout the paper there was vague wording like "entity level" that made it difficult to understand exactly what was going on and what the challenges were.
- I'd suggest formalizing the task setting in a similar way to how POSG was formalized; I found that part easy to read
- I would suggest adding something formal about the game's incentives (it seemed like these games were both collaborative and competitive at the same time, maybe because you're learning a policy that's part of a team that's acting against another team?)
- The representation of subgoals and subgoal division was also unclear; formalization would help here.
- The word "opponent" modeling is confusing if some of the other entities are collaborating with one another, and aren't opponents.

Originality: I'm not very familiar with multi-agent RL work, so I am not sure how novel it is from that side of things. From language-based RL, the setting appears to be rather conventional, except for the multi-agent setting (which is the novelty of this paper).

**Strength And Weaknesses:**

Strengths:
- The use of "opponent" modeling as an auxiliary loss is intriguing (although I don't know how conventional it is in multi-agent RL already).
- Description of POSG is clear (Section 3)
- Performance of proposed method significantly outperforms existing methods
- Experiments are relatively thorough

Weaknesses:
- See below for clarity concerns.
- I'd like more discussion on what is functionally different between the two environments that make both of them good to study.
- I found it a bit confusing that EnDi enabling agents to individually model subgoals also helps the agents coordinate. By modeling subgoal division, are the agents explicitly predicting how all of the other agents will act next? Does this mean it's possible for two collaborative agents to fail to complete a subgoal under the "assumption" the other will complete it, e.g., if they are both equally far away from the entity relevant to that subgoal?

Some questions:
- The opponent modeling head actually predicts multiple actions, corresponding to all of the other agents, right?
- In Eq. 1, does each agent have access to the observations of the other agents, or is it predicting others' actions based on its own partial observation of the world?
- How were the stages for the curriculum learning chosen?
- Supervised loss ablation (EnDi-sup) is equivalent to ablating the opponent modeling, right?
- Why evaluate QMIX, rather than ablating language as much as possible in EnDi? E.g., by removing access to the manual representation z?

**Summary Of The Paper:**

This paper proposes a new method for improving generalization in multi-agent games, where game manuals describing entity and environment dynamics are provided, and high-level goals are specified in natural language. The proposed method learns a policy for each entity, trained with reinforcement learning, with an auxiliary supervised loss on predicting the dynamics of other agents in the environment. They modify two existing similar environments to be multi-agent (MESSENGER and RTFM) and evaluate with several comparisons to other methods and ablations.

**Summary Of The Review:**

This paper was interesting, and I think the contribution is potentially useful, but I did find the description of the task itself to be relatively informal to the point of being somewhat difficult to understand the contributions. I am also not deeply familiar with multi-agent work, so cannot evaluate it thoroughly from that point of view.

---

> ### Author Response · Authors · 2022-11-18
> **Responses to Reviewer imbM**
>
> We appreciate the reviewer for your valuable and inspiring comments on our work, which have greatly helped to improve the quality of our paper.
>
> > I'd like more discussion on what is functionally different between the two environments that make both of them good to study.
>
> Firstly, they are the only two latest papers that focus on the generalization ability of policies to new environments based on language. No further environments are there available in this research field.
>
> A key challenge in RTFM is multi-modal multi-step reasoning with texts associated with multiple entities, while a key challenge in MESSENGER is the adversarial train-evaluation split without prior entity-text grounding.
>
> > I found it a bit confusing that EnDi enabling agents to individually model subgoals also helps the agents coordinate. By modeling subgoal division, are the agents explicitly predicting how all of the other agents will act next? Does this mean it's possible for two collaborative agents to fail to complete a subgoal under the "assumption" the other will complete it, e.g., if they are both equally far away from the entity relevant to that subgoal?
>
> We mentioned in Sec 4.1 on the subgoal division module that we use Gumbel softmax to generate binary masks to avoid this problem, which ensures that agents do not try to complete the same subgoal at the same time.
>
> > The opponent modeling head actually predicts multiple actions, corresponding to all of the other agents, right?
>
> Yes, the opponent modeling head predicts the actions of all the other agents. We will change the terminology to agent modeling to avoid misunderstanding.
>
> > In Eq. 1, does each agent have access to the observations of the other agents, or is it predicting others' actions based on its own partial observation of the world?
>
> No, agents cannot access the observations of others.
>
> > How were the stages for the curriculum learning chosen?
>
> We follow the original single-agent version RTFM and MESSENGER for most curriculum settings.
> However, we add stage 1 in multi-RTFM to make training easier. Note that the curriculum settings are the same for all the baselines.
>
> > Supervised loss ablation (EnDi-sup) is equivalent to ablating the opponent modeling, right?
>
> Yes
>
> > Why evaluate QMIX, rather than ablating language as much as possible in EnDi? E.g., by removing access to the manual representation z?
>
> In previous works, such as RTFM and MESSENGER, the language ablation has been done thoroughly and they reach a conclusion that language-free methods cannot exhibit a promising generalization. This can be regarded as common consent in this research field.
>
> All the proposed methods are designed for language input. If there is no language input, there is no doubt that QMIX is more suitable and can achieve higher rewards than EnDi in non-language tasks. We aim to provide a general conclusion that even SOTA language-free methods cannot exhibit a promising generalization.
>
> > Experiments are mostly thorough. I would have liked to see a bit of error analysis to see what kind of mistakes are still being made in this environment.
>
> We have added error analysis in the appendix.
>
> > Throughout the paper there was vague wording like "entity level" that made it difficult to understand exactly what was going on and what the challenges were.
>
> "Entity level" means that all the proposed methods are based on the entities in the environment. Language grounding aims to bind the language descriptions to the entities. The subgoal mask is used to ignore irrelevant entities.
>
> > - I'd suggest formalizing the task setting in a similar way to how POSG was formalized; I found that part easy to read
> > - I would suggest adding something formal about the game's incentives (it seemed like these games were both collaborative and competitive at the same time, maybe because you're learning a policy that's part of a team that's acting against another team?)
> > - The representation of subgoals and subgoal division was also unclear; formalization would help here.
> > - The word "opponent" modeling is confusing if some of the other entities are collaborating with one another, and aren't opponents.
>
> Thank you for your insightful advice. We will revise it in the next version.

---

> > ### Comment · Reviewer_imbM · 2022-12-13
> > **thanks for response**
> >
> > Thank you for your response to my review. After reading the other reviews and through some discussion, I agree with other reviewers' concerns about MARL evaluation and generalization. My main suggestion would still be to clarify the task and proposed method. I usually find papers easiest to read when I could clearly implement pseudocode or an API representing the proposed task / method without asking the authors for clarification. I felt there were too many details missing in the setup / proposed approach here for me to be able to understand it to that level.

---

### Official Review · Reviewer_affK · 2022-10-25

**Confidence:** 4
**Correctness:** 3
**Technical Novelty And Significance:** 3
**Empirical Novelty And Significance:** 3
**Recommendation:** 6

**Clarity, Quality, Novelty And Reproducibility:**

### Clarity:
While some clarification is needed, the paper is generally well-written and easy to follow

### Quality:
The claim is mostly supported by the experiments. However, more comprehensive experiments may be needed. Please see the weakness / questions section for details.

### Novelty:
The paper formulated and addressed the important language grounding problem for MARL. The reviewer found the setting and subgoal division method interesting. However, the reviewer has some concern regarding the generality of the proposed approach.  Please see the weakness / questions section for details.

### Reproducibility:
The description and the hyper-parameter aew clear. However, no code for the approach and the environment is provided. Without the code of the environment, it would be hard to reproduce the results.


**Strength And Weaknesses:**

### Strength:
1. Language grounding for multi-agent settings is an important area which draws less attention. The reviewer thinks this paper took a first step toward this goal.

2. The reviewer found the proposed sub-goal division, which based on opponent modeling, simple yet interesting. The experimental results also demonstrate its effectiveness on two MARL environments.

3. The paper is well-written and easy to follow.

### Weakness / Questions:

1. To get $X^{others}_{goal}$, joint position feature of all other agents ($l_{-i}$) is needed.  Isn’t it a privileged knowledge that violates the partially observable assumption discussed in Section3 and the ‘decentralized execution’ of general multi-agent settings?

2. Are the binary entity masks on the full grid? Or is it only on each agent’s visual range (local observation)? In addition, what is the visual range of each agent in the experiments?

3. It seems that the opponent modeling (Eq. 1) is a critical component for avoiding sub-goal conflict. However, opponent modeling itself is a challenging problem. It is unclear how the proposed method ensures the modeling results are good and thus lead to good subgoal division? In addition, Suppose agent A never occurs in another agent’s, e.g. agent B, visual range. How could agent B model agent A’s behavior correctly?  If the agent is not able to model others’ behaviors accurately, how would it impact the quality of the binary entity mask?

4. It is unclear how many agents each task. It seems that there are only two controlled agents in a task. The experimental section could be more convincing if there are more controlled agents in the tasks.

5. In p. 8, the paper explains Fig. 3: “ it chooses to go down to the target monster, leaving the upper target monster to Agent1…”. Given the upper monster is left for agent 1, why is the upper monster assigned to the mask of agent 2?

6. Is the proposed sub-goal approach applicable to environments that are not grid-world based?


**Summary Of The Paper:**

This paper studies using natural language as instructions and manuals for multi-agent reinforcement learning (MARL). The problem is challenging because the agents need to ground the language and consider other agents’ behavior at the same time. This paper proposed a language-grounding framework, which is referred to as ‘entity divider (EnDi)’, for MARL. Based on opponent modeling, the proposed approach enables agents to divide subgoals at entity level and act with the environment accordingly. The proposed EnDi is evaluated on multi-agent RTFM and multi-agent MESSENGER which are augmented from the popular RTFM and MESSENGER environment. The experimental results show that the proposed approach outperforms baselines and is able to generalize to unseen tasks.


**Summary Of The Review:**

In summary, the reviewer found the problem and the proposed approach interesting. However,  it is unclear if the proposed approach is applicable to more complex environments, e.g. more agents, non-grid-world tasks, and limited visual range. Further clarification and revision are needed.

---

> ### Author Response · Authors · 2022-11-18
> **Responses to Reviewer affK**
>
> We greatly appreciate your valuable and constructive comments on our work, which have greatly contributed to the quality of our papers.
>
> > The partial observable issues.
>
> Importantly, we strictly follow the observation settings of previous works, RTFM and MESSENGER. That is, we adopt the full grid vision range as default. However, the partially observable setting is not only reflected in the vision range. In MESSENGER, the agent cannot observe the text reference of each entity in the grid world, which is important for entity identification. Moreover, the agent cannot know the role of each entity since each game each entity could be assigned a new role. Therefore, memorizing the role based on entity symbols is impossible. In addition, in multi-agent settings, agents can hold entities as weapons, but they cannot observe the possession and other properties of other agents, such as health and damage. Since without the language, it is unlike for the agent to learn a good policy and we claim the observation is partial observation as previous works. In this line of research, we focus on a different view of partial observability.
>
> The position information is also available in previous work, such as RTFM.
>
> Yes, opponent modeling itself is a challenging problem. We jointly optimize the supervised loss and reinforcement loss. The decreased supervised loss can reflect the modeling results. If the supervised loss does not decrease, EnDi should have the same performance as EnDi-sup. However, the ablation results show if we remove the supervised loss, the performance decreases. Note that we have shown the simplest opponent modeling can help improve the performance. There is no reason that other complex opponent modeling methods cannot help.
>
> For a small vision range, if an agent cannot see certain agents, it will freeze the policy heads for those agents. The agent would not consider the unobservable agents in the binary mask. Intuitively, agents at a distance would exert limited influence on the decision-making of self-agent. That is why agents only communicate with nearby agents in MARL communication.
>
> > It is unclear how many agents each task. It seems that there are only two controlled agents in a task. The experimental section could be more convincing if there are more controlled agents in the tasks.
>
> There are two agents in the task. Note that EnDi can be applied to any scenario with n agent. In Section 5.2 of the Extension (page 8), we also experimentally validate the case of three agents, and the results show that EnDi can bring improvement for scenarios with more than two agents.
>
> > In p. 8, the paper explains Fig. 3: “ it chooses to go down to the target monster, leaving the upper target monster to Agent1…”. Given the upper monster is left for agent 1, why is the upper monster assigned to the mask of agent 2?
>
> The reviewer may have a misunderstanding in this paragraph, the situation described here is that the upper monster is assigned to agent 1 instead of agent 2.
>
> > Is the proposed sub-goal approach applicable to environments that are not grid-world based?
>
> EnDi's subgoal division module is not limited to the grid-world environments. If language grounding with entities is possible in other environments, EnDi can take language-based representation as input and output a binary mask to remove the irrelevant entities from any kind of observation. In addition, graph methods can be used to connect all the entities and agents to form another form of grid world.
>
> > The description and the hyper-parameter are clear. However, no code for the approach and the environment is provided. Without the code of the environment, it would be hard to reproduce the results.
>
> We promise in the Open-Review platform that we will open-source the code once this paper is accepted.

---

### Official Review · Reviewer_dLxy · 2022-10-25

**Confidence:** 2
**Correctness:** 3
**Technical Novelty And Significance:** 2
**Empirical Novelty And Significance:** 2
**Recommendation:** 6

**Clarity, Quality, Novelty And Reproducibility:**

The clarity of the abstract and the main body has room for improvement.
The quality of the paper looks good. Experiment settings are described clearly to a certain extent.

**Strength And Weaknesses:**

Strength
The experiment shows that their proposal is working better than the baselines.

Weakness
The generality of the proposed model in broader MARL studies is not apparent.

**Summary Of The Paper:**

The authors address the challenges of grounding language in generalization for unseen dynamics in multi-agent settings. They propose a new framework for language grounding called entity divider (EnDi).
EnDi enables agents to independently learn subgoal division at the entity level and act in the environment. This division is regularized by the opponent
modeling to avoid subgoal conflicts and promote coordinated strategies.
The authors also conducted experiments to show that EnDi outperformed existing language-based methods in all tasks by a large margin.

**Summary Of The Review:**

In summary, this is an interesting paper that provides a new method. However, the description and style may have room for improvement. It is better to make the abstract and conclusion more detailed and explicit.

---

> ### Author Response · Authors · 2022-11-18
> **Responses to Reviewer dLxy**
>
> We are very grateful for your recognition of our work and your valuable suggestions for our work.
>
> To allay your concerns about the generality of the proposed model, we will again explain EnDi's contribution to Generalization. As the key idea we emphasized in the paper, language grounding is invariant to a new distribution of dynamics or tasks, making the generalization more reliable. We jointly ground the text manual and the language-based goal (goal instruction) to environment entities, each entity has been associated with explicit dynamic rules and relationships with the goal state. This design is compatible with any game environment of the same type that has a text manual. As for the broader MARL, we need to clarify that even for environments without text manuals, we can still add some general text descriptions to these environments to be compatible with EnDi and take advantage of the improvements that EnDi brings.

---

> > ### Comment · Reviewer_dLxy · 2022-12-09
> > **Thanks for your response**
> >
> > Thanks for your response.
> > Reading your response to other reviewers' comments, I understand that you believe the method can be generalized to the n-agent and non-grid MARL settings.
> > I hope you will be able to provide evidence in your future work. Thanks again for submitting a nice paper.

---

### Official Review · Reviewer_xDyS · 2022-10-31

**Confidence:** 4
**Correctness:** 3
**Technical Novelty And Significance:** 3
**Empirical Novelty And Significance:** 3
**Recommendation:** 3

**Clarity, Quality, Novelty And Reproducibility:**

To my knowledge, there is no prior work in multi-agent language-assisted RL, so the setting is novel. I am also unaware of similar approaches to subgoal assignments in MARL, though I am less familiar with this literature.

I found the paper difficult to follow in places with some important details left unclear or insufficiently motivated. The following parts of the paper need to be clarified before the paper is ready for publication:
-  the function Ground(.) that is mentioned across Section 4.1: is this always the same function, is there parameter sharing within or between agents?
- what are the dimensions of mask $m_i$? does it mask just the entities, positions or also features of the entities?
- in Section 4.1 subgoal division module, what exactly is the target distribution $p_e^*$ and how do you ensure it has a relatively low variance? Clarify that |e| refers to the number of entities (if I'm interpreting this correctly)
- in Section 5.2 performance, is $|e_{i}|$ number of entities covered by mask of agent $i$? What is the form of these losses when there are more than 2 agents?
- which version of the RTFM task is shown in Figure 3?

The paper also needs a couple more passes to fix various grammatical and writing errors.

In addition, I'd suggest the following (less important) changes to improve clarity:
- add examples from the task to p5 and p6 of introduction
- explain what motivates the choice of curriculum in Section 5.1, how did you decide on this particular curriculum and how well do the models perform without it
- the visualizations of architecture in Fig 2 and 5 are hard to understand on their own, the figure captions should be expanded to make the figures understandable without the referrals to various sections and prior work
- the usage of the term opponent is confusing, given the agents are trained in a cooperative setting
- separate the ablation experiments into another table (from Table 1); it is hard to keep track of different variants of the model
- explain what is meant by rationality of the subgoal division (sec 4.1 and 5.2) (maybe a better term is complementary/uniform coverage of entities?)
- explain what is the shaded in Fig 4
- the term natural language typically refers to the language generated by humans, hence I'd refer to language in RTFM as simply templated language
- Table 1-3: explain the table column names in the caption
- in section 5.3 multi-agent RTFM, S5-Eval(new), clarify that all monsters and modifiers are novel
- add references to the Appendix; there is often important clarifying information in appendix, but it is not referred to when applicable

**Strength And Weaknesses:**

I appreciated that the setting addressed in this paper is novel, in general, language-assisted RL is not well-studied and it's good to see more work in this area. The results are certainly promising, the proposed method strongly outperforms independently trained agents, with particularly interesting results in generalization to unseen entities / modifiers and grid-sizes. The proposed architecture overall makes sense.

However, this work is not ready for publication due to a lack of comparison to standard MARL baselines, methodological issues, and a lack of clarity about the method (see Clarity section).

The authors do not specify well which grounding-related problems are introduced by the addition of multi-agent dynamics to language-assisted RL beyond standard problems of MARL (i.e. problems related to incorporating language or attending to manual; instead of just increased non-stationarity and learning coordination strategies). The authors touch on this question in p2 of the Introduction, however, this needs to be elaborated on better. As it stands, the proposed method could be used in most MARL benchmarks and standard MARL methods (such as QMIX [1], MAPPO [2]) could be combined with architectures more suitable for grounding such as EMMA and text2pi when the task demands language grounding. So the authors could also examine the proposed architecture in standard non-language MARL tasks; but in either case, the baselines need to include standard MARL methods. While one of the experiments does contain a comparison to QMIX (Section 5.3), this is only in a language-free setting (i.e. using QMIX with a standard architecture that is not suitable for language grounding). If there is a reason why standard MARL methods can not be combined with grounding architectures, the authors need to elaborate on it in the paper.

If I'm interpreting this correctly, the results are reported over environment random seeds, but not training seeds. In section B.2 of the Appendix, the authors say that the validation games are used to select the highest-performing parameters instead of hyperparameters, with the reported results corresponding to this best-performing model on the random selections of test environments. This is very non-standard in RL for a variety of reasons [3], including not giving a sense of how stable the training is. The authors also do not seem to perform any hyperparameter search over either the proposed model or the baselines, which is a problem since the default baseline hyperparameters were optimized on another version of the environment. My suggestion is to perform a hyperparameter search, use the validation performance for hyperparameter selection, then report the results over both training and environment seeds for the best hyperparameters (but not the model).

The generalization results (Table 2 & 3) do not report baseline performance.

Lastly, there is only one experiment with more than 2 agents (it's only 3 agents, and the experiment is on the easiest version of one of the environments). The performance gap is much smaller there, with the proposed model barely outperforming the non-MARL baseline (why is this the case?). This sheds doubt on whether the proposed method works as well with more than 2 agents and in more challenging environments.

[1] https://arxiv.org/abs/1803.11485
[2] https://arxiv.org/abs/2103.01955
[3] https://arxiv.org/abs/1709.06560

**Summary Of The Paper:**

The paper addresses the problem of collaborative MARL with language for specifying instructions and environmental dynamics, focusing on the problem of distributing subgoals among multiple agents. The approach proposed in the paper builds upon the grounding architecture from Zhong et al, 2019 and Hanjie et al. 2021, with two main architectural changes meant to enable individual agents to focus on different subgoals: (a) complementary masking over observations, leading to different agents attending to different entities in the observation; and (b) modeling of other co-operating agents as an auxiliary objective.

In the experiments on multi-agent versions of Messenger (Hanjie et al, 2021) and RTFM (Zhong et al, 2019), the proposed architecture outperforms independently trained agents (without the modifications to the architecture), also demonstrating generalization to completely new monsters / modifiers and increased grid size.

**Summary Of The Review:**

The paper proposes several architectural changes to enable subgoal division in collaborative MARL with language for specifying instructions and environmental dynamics. The results with two agents are promising, with the proposed method significantly outperforming independently trained agents with a non-modified grounding architecture. However, the paper is not ready to be accepted due to a lack of comparison to standard MARL baselines, methodological issues, and a lack of clarity about the method.

---

> ### Author Response · Authors · 2022-11-18
> **Responses to Reviewer xDyS - Part 1**
>
> To begin with, we thank the reviewer for the carefully reviewing and insightful advice.
>
> >The authors do not specify well which grounding-related problems are introduced by the addition of multi-agent dynamics to language-assisted RL beyond standard problems of MARL
>
> We want to address the generalization problem with the help of language grounding. During the language grounding process, agents need to learn the meaning of the language manual (e.g., how each entity behaves) through agent-entity interactions. If the agents can understand the dynamics of the environment from the language manual, generalization of policies to a new environment is possible.
>
> For grounding-related problems, the number of entities might increase in multi-agent settings since more complex goals are required. Normally, agents need to bind each sentence to certain agent-entity interactions. With the increasing number of entities, so is the search space for language grounding. We aim to address this problem by ignoring some irrelevant entities in the training process dynamically.
>
> As for non-stationarity in MARL, it is not relevant to language grounding. Note that we focus on the generalization of policies by relating the dynamics of the environment with the text manual that describes how entities behave. However, the dynamics in the multi-agent setting also include the actions of other agents and the language manual cannot reflect the policies of others, especially in a new environment. We address this point by adopting opponent modeling and capturing it at the entity level after language grounding.
>
> Based on our motivation, the addressed problems mentioned above are orthogonal to backbone algorithms. In other words, MAPPO or QMIX cannot help reduce the search space of the grounding process and consider others' actions in a new environment.
>
> We doubt the necessity of combining extra MARL methods. Note that EnDi is respectively built on IMPALA and PPO in multi-RTFM and multi-MESSENGER following the original papers. All the implementations are for a fair comparison. We want to demonstrate the effectiveness of our method by removing the influence of the backbone. Even if MAPPO+EMMA or MAPPO+txt2pi can get better performance than EnDi, it is not a fair comparison and the performance gain could come from the superiority of MAPPO over PPO or IMPALA.
>
> As for examining standard non-language MARL tasks, our proposed method is based on language input. If there is no language input, agents cannot know the role of each entity. Note that in each game the entity can be assigned a new role. The agent cannot memorize the roles when the distractor entity exists. Since our method is only designed for language-based RL, the performance of EnDi in non-language tasks does not affect the conclusion we have achieved in this paper.
>
> >the authors say that the validation games are used to select the highest-performing parameters instead of hyperparameters.
>
> Firstly, the environment and training seed are both changed.
>
> The original paper of MESSENGER adopts this training procedure, we just strictly follow it for a fair comparison. We are worried to be questioned if not.
>
> However, we also run the experiments following the settings the reviewer mentioned and we have updated them in the Appendix. The results show a similar performance with a slight drop comparing the original version.
>
> >The generalization results (Table 2 & 3) do not report baseline performance.
>
> The performance of baselines before generalization is relatively low around 2-5%, which means the policies are not converged and are totally useless. The generalization results of such policies are meaningless.
>
> >Lastly, there is only one experiment with more than 2 agents (it's only 3 agents, and the experiment is on the easiest version of one of the environments).
>
> The extension experiment is just to illustrate how to implement our method in complex settings with more than two agents.
>
> In the two-agent settings, the result is 100% vs 99%. Therefore, we would say the performance gap in 3-agent settings is not small.
>
> We admit in the current experiment setting, we cannot fully convince the reviewer of whether the proposed method can perform well in more challenging environments. However, compared with the previous works, our work moves one step forward, firstly investigating language grounding at the entity level in multi-agent settings. We have demonstrated the huge performance gap between single-agent methods and EnDi in the two-agent settings. We believe with the insight we provided, future work can do much better.

---

> > ### Author Response · Authors · 2022-11-18
> > **Responses to Reviewer xDyS - Part 2**
> >
> > >Other clarifications
> > >> the function Ground(.) that is mentioned across Section 4.1: is this always the same function, is there parameter sharing within or between agents?
> >
> > No, the parameter is not shared
> >
> > >>what are the dimensions of mask mi? does it mask just the entities, positions or also features of the entities?
> >
> > The dimension of the binary mask is $H \times W$. It masks the entities.
> >
> > >>in Section 4.1 subgoal division module, what exactly is the target distribution pe∗ and how do you ensure it has a relatively low variance? Clarify that |e| refers to the number of entities (if I'm interpreting this correctly)
> >
> > Intuitively, we are willing to observe that all agents make contributions to the goal. If the variance is high, It is hard to guarantee that all agents can contribute.
> >
> > >>in Section 5.2 performance, is |ei| number of entities covered by mask of agent i? What is the form of these losses when there are more than 2 agents?
> >
> > We mentioned this part in the 5.2 performance part. Yes, |ei| is the number of entities covered by the mask. The form is ||ei| − |e−i|/(n − 1)|, where n is the number of agents.
> >
> > >>which version of the RTFM task is shown in Figure 3?
> >
> > In the caption of Figure 3, we mentioned that it is the multi-agent RTFM. The task is to defeat all the entities belonging to the order of the forest.

---

> > > ### Author Response · Authors · 2022-11-24
> > > **Message to Reviewer xDyS**
> > >
> > > From your comments, we can tell that you have a deep understanding of multi-agent learning and that you have read our paper in detail. Your comments helped us correct errors in our manuscript and turn it into a better shape which we appreciate. We would really appreciate it if you could read our responses and amendments and reconsider your assessment. Moreover, we hope more tolerance can be given since you mentioned that our method is novel and interesting with promising results and our work indeed moves one step forward in multi-agent settings.

---

> > > ### Comment · Reviewer_xDyS · 2022-12-09
> > > **Appreciate the improvements, further notes**
> > >
> > > I'm glad the authors found the feedback helpful! I appreciate the clarifications added to the paper, as well as the addition of examples of failure modes in Appendix F and experiments without the model selection in Appendix E.
> > >
> > > The setting is interesting and the results look very promising. That being said, after reading the other reviews, the author's response, and the updated version of the paper, I still don't think the paper is ready to be accepted yet:
> > >
> > > 1. I am not convinced that the aforementioned MARL baselines are not warranted in the case. As far as I can say, the two main algorithmic contributions ((a) agent-dependent observation masking and (b) agent modelling) are mostly MARL solutions orthogonal to the problem of grounding, so it is hard to disentangle to what extent the reported improvements are due to addition of (a) & (b) vs. combining grounding architectures like text2pi with MARL solutions. The kind of baseline I'm proposing here would use a MARL objective/architecture on top of text2pi state embedding.
> > > 2. for a proposed MARL method, it is important to know how the method performs with the increased number of agents, hence there should be more experiments with three or more agents. Running those experiments is even more important here given that the only available experiment with >2 agents indicates that the performance gap might be much smaller as the number of agents increases (1% advantage in one experiment is not very convincing).
> > > 3. the clarity could still be significantly improved (e.g. see direct response below; in Appendix E, without the context of this discussion, it is unclear what is meant by validation).
> > >
> > >
> > > Minor:
> > > - since writing the review, I became aware of the following related work. It might be a particularly strong and interesting baseline (as described in (1)) for comparison:
> > > https://arxiv.org/abs/2006.04222
> > > - I would still argue that you need to report in the main body of the paper (not just the appendix) the generalization performance averaged over all trained models (with the same hyperparameters), and not just the model selected based on validation performance. Validation is meant for the selection of hyperparameters, not parameters.
> > >
> > >
> > > Other direct responses:
> > >
> > > > Intuitively, we are willing to observe that all agents make contributions to the goal. If the variance is high, It is hard to guarantee that all agents can contribute.
> > >
> > > The question was how was the target distribution $p^*_e$ defined, i.e. what is its mathematical form? Paragraph 5 in Section 4.1 only specifies the mean of the target distribution.
> > >
> > > > In the caption of Figure 3, we mentioned that it is the multi-agent RTFM. The task is to defeat all the entities belonging to the order of the forest.
> > >
> > > By task I meant what is referred to in the paper as stages, i.e. which task difficulty level / stage for RTFM/Messenger is visualized there?

---

> > > > ### Author Response · Authors · 2022-12-10
> > > > **Response to Reviewer xDyS**
> > > >
> > > > Many thanks for responses.
> > > >
> > > > We believe there are some big misunderstandings and we do hope the reviewer can have a second thought.
> > > >
> > > > First, we claim the backbone algorithms, such as MAAPO, QMIX, PPO, or REFIL (it can be seen as entities-based QMIX and it focuses on multi-task rather than generalization we consider, https://arxiv.org/abs/2006.04222) are orthogonal to the problem of grounding. The change of backbone cannot reduce the search space of the grounding process or consider others' actions in a new environment. But our proposed methods, i.e., (a) agent-dependent observation masking and (b) agent modeling, are **highly relevant to the grounding problem**. (a) is used to reduce the number of agent-entity interactions, while (b) is used to focus on the right entities even in the new environment. With our proposed method, language grounding can be efficient.
> > > >
> > > > Our ablation studies are to verify backbone + language grounding + (a) or (b) can outperform the backbone + language grounding. Since the backbone algorithms are orthogonal to the groundings, we think our ablation studies can verify the effectiveness of each component. In the language-based RL, we should focus more on the generalization brought by grounding methods and the backbone algorithms are only tools.
> > > >
> > > > Second, **we mentioned in the two-agent settings, the result is 100% vs 99%, as Table 1.** However, **in the three-agent settings, the performance gap is much larger (around 88% vs 82%).** Intuitively, as the number of agents increases, reducing the search space becomes more and more important. The single-agent methods suffer in the two-agent settings (2%-5% win rate in the final stages), thus they cannot have good performance in the more complex settings.
> > > >
> > > > Note that the language-based RL is not easy to train. Otherwise, previous works wouldn’t struggle in single-agent grid world and no curriculum learning is needed. The experiment results show the agents must go through a long process of exploration to learn to ground language, showing the large gap between the single-agent and multi-agent settings.
> > > >
> > > > Third, due to the page limit (9 pages), it is hard for us to add new results without removing the original results since we think following the settings of previous work is also important. However, we agree with the reviewer's advice and we will add more discussion and add them to the main body in the revised version.
> > > >
> > > > Others issues: The target distribution has zero variance and the final stages are visualized in Figure 3.
> > > >
> > > > Lastly, we noticed that the reviewer also mentioned the setting is interesting and the results look very promising. All works have limitations, but we do believe a paper with new insight and promising results should be encouraged.

---

> > > > > ### Comment · Reviewer_xDyS · 2022-12-12
> > > > > **Quick final response**
> > > > >
> > > > > > First, we claim the backbone algorithms, such as MAAPO, QMIX, PPO, or REFIL (it can be seen as entities-based QMIX and it focuses on multi-task rather than generalization we consider, https://arxiv.org/abs/2006.04222) are orthogonal to the problem of grounding. The change of backbone cannot reduce the search space of the grounding process or consider others' actions in a new environment. But our proposed methods, i.e., (a) agent-dependent observation masking and (b) agent modeling, are highly relevant to the grounding problem. (a) is used to reduce the number of agent-entity interactions, while (b) is used to focus on the right entities even in the new environment. With our proposed method, language grounding can be efficient.
> > > > >
> > > > > I think this needs to be demonstrated empirically by running the experiment I'm suggesting.
> > > > >
> > > > > > Second, we mentioned in the two-agent settings, the result is 100% vs 99%, as Table 1. However, in the three-agent settings, the performance gap is much larger (around 88% vs 82%). Intuitively, as the number of agents increases, reducing the search space becomes more and more important. The single-agent methods suffer in the two-agent settings (2%-5% win rate in the final stages), thus they cannot have good performance in the more complex settings.
> > > > >
> > > > > My apologies, the 1% figure was taken from your first response which I now see refers to the experiment with two agents. That being said, it is still only one experiment with 3 agents, and as far as I can tell, the uncertainty measures in Figure 4 are quite overlapping.
> > > > >
> > > > > > Third, due to the page limit (9 pages), it is hard for us to add new results without removing the original results since we think following the settings of previous work is also important.
> > > > >
> > > > > Note that any new results can easily be added to the appendix.

---

> > > > > > ### Author Response · Authors · 2022-12-12
> > > > > > **Final Response to Reviewer xDyS**
> > > > > >
> > > > > > >>This needs to be demonstrated empirically.
> > > > > >
> > > > > > Again, this work is not investigating the relationship between different backbones and language grounding. Since there is no theoretical basis provided by the reviewer, we still argue this is beyond the scope of this paper.
> > > > > >
> > > > > > >>The new results could easily have been added to the appendix.
> > > > > >
> > > > > > As the reviewer mentioned, Appendix E is new results. We only explained why we didn’t move it to the main body.
> > > > > >
> > > > > > Last, although we did not agree with some opinions of the reviewer, we still thank the reviewer for providing some helpful reviews.

---

### Author Response · Authors · 2022-11-18
**Uploading a Revised Version**

We have uploaded a revised version of the manuscript which addresses the concerns of reviewers, including the clarification issues mentioned by reviewers, the results of multi-Messenger under standard RL settings mentioned by reviewer xDys, and the error analysis mentioned by reviewer imbM.

---

### Decision · Program_Chairs · 2023-01-20

**Decision:**

Reject

**Justification For Why Not Higher Score:**

See reviewer discussion above.

**Justification For Why Not Lower Score:**

N/A

**Metareview: Summary, Strengths And Weaknesses:**

This paper tackles the problem of language grounding in a multi-agent RL environment, where language is used to specify instructions and the environment dynamics. Building on two existing environments, the paper creates multi-agent versions with subgoals distributed among agents. It then proposes a model that enables each agent to focus on different subgoals through complementary masking and modeling other co-operating agents. Experiments on both environments show that the proposed multi-agent method outperforms independently trained agents.

The reviewers appreciated the novelty of the setup and the importance of the language grounding task. However, there were concerns around 1) the clarity of the paper, where several important details on the approach were either unclear or not present, 2) the lack of empirical evidence with several simpler MARL baselines combined with grounding baselines from prior work such as txt2pi or EMMA, and 3) more results with 3+ agents. The author response did not convince the reviewers on these weaknesses and the reviewers agreed in the discussion that the paper could use further work in these areas to make it stronger.


**Summary Of Ac-Reviewer Meeting:**

The discussion was between 3 reviewers, including two with highly divergent scores. Reviewer affK was not present.

Positives for the paper:
R1: Results are promising, effect sizes are significant, really promising initial results,
R2: results are good, setting is novel and interesting.
R3: new setting, well motivated, use of opponent modeling as aux loss, results are good.

Weaknesses:
1) The paper's technical parts are not clear. It will not be easy to reproduce the method. ("formalization of the task was lacking, hard to understand the assumptions made in the work, wording was vague in place")
2) Comparison with simpler MARL+grounding baselines
3) Results with 3+ agents is lacking. Multi-agent does not only mean 2 agents.

The reviewers came to a consensus that while the paper has good ideas, several important questions remain unanswered, hence recommend a weak reject.